# A Contemporary Review of Antiplatelet Therapies in Current Clinical Practice

**DOI:** 10.3390/ijms241311132

**Published:** 2023-07-05

**Authors:** Sacchin Arockiam, Brittany Staniforth, Sacha Kepreotis, Annette Maznyczka, Heerajnarain Bulluck

**Affiliations:** 1Yorkshire Heart Centre, Leeds General Infirmary, Leeds Teaching Hospitals NHS Trust, Leeds LS2 3AX, UK; 2Leeds Institute of Cardiovascular and Metabolic Medicine, University of Leeds, Leeds LS2 9JT, UK

**Keywords:** antiplatelet therapy, atherosclerotic vascular disease, ischaemic heart disease, acute coronary syndrome, coronary artery bypass graft surgery, stroke, peripheral vascular disease, transcatheter aortic valve implantation, coronavirus 2019

## Abstract

Antiplatelet therapy plays a crucial role in a number of cardiovascular disorders. We currently have a range of antiplatelet agents in our armamentarium. In this review, we aim to summarise the common antiplatelet agents currently available, and their use in clinic practice. We not only highlight recent trials exploring antiplatelet therapy in atherosclerotic cardiovascular disease, but also in trials related to transcatheter aortic valve implantation and coronavirus disease 2019. Inevitably, the antithrombotic benefits of these drugs are accompanied by an increase in bleeding complications. Therefore, an individualised approach to weighing each patient’s thrombotic risk versus bleeding risk is imperative, in order to improve clinical outcomes.

## 1. Introduction

Under normal physiological conditions, platelets are important to maintain normal haemostasis. However, in certain situations, platelet adhesion, activation and aggregation also play a crucial part in the pathophysiology of thrombotic complications within the cardiovascular system [1].

As such, antiplatelet therapy has a well-established role for secondary prevention in a range of cardiovascular disorders, ranging from ischaemic heart disease (IHD) [2,3] to stroke [4] and peripheral artery disease (PAD) [5]. Currently, we have both oral and intravenous antiplatelet agents in our armamentarium to treat a wide range of conditions. Inevitably, the antithrombotic benefits of these antiplatelet agents are accompanied by an increase in bleeding complications. Therefore, an individualised approach to weighing each patient’s thrombotic risk versus bleeding risk is imperative.

Over the past few years, there have been several randomised controlled trials that have informed our practice, and risk scores have been developed to assess patients’ thrombotic versus bleeding risk [6,7]. In this review, we aim to summarise the common antiplatelet agents currently available, and their use in clinic practice. We not only highlight recent trials exploring antiplatelet therapy in atherosclerotic cardiovascular disease (ASCVD), but also in trials related to transcatheter aortic valve implantation (TAVI) and coronavirus disease 2019 (COVID-19). Further details of the trials discussed in each section below are available in the Appendix A.

## 2. Antiplatelet Agents in Current Clinical Practice

There are several oral and intravenous antiplatelet agents that are available for use in current clinical practice, and they are outlined below according to their mechanism of action to achieve platelet inhibition (Table 1).

Aspirin is the most widely used antiplatelet agent; it exerts its antiplatelet property by irreversibly inhibiting the cyclooxygenase-1 (COX-1) enzyme present in the platelets. This in turn blocks thromboxane A2 production, which is a potent platelet activator [8]. Aspirin also blocks the COX-2 enzyme, thereby exhibiting its anti-inflammatory property. Aspirin has also been shown to inhibit prostaglandins (e.g., PGE2 and PGI2), which has a proinflammatory effect. Earlier researchers found that lower doses of aspirin had greater inhibitory effects on thromboxane A2 than PGI2 metabolites. However, they found that platelet function inhibition was not maximal at lower aspirin doses, and that aspirin at daily doses >80 mg caused substantial inhibition of endogenous PGI2 production. A daily dose of 75–100 mg of aspirin is usually sufficient to achieve maximal antithrombotic effect. Higher doses of aspirin do not confer any additional antithrombotic benefit, whilst increasing the risk of bleeding [9]. The terms “aspirin resistance” or aspirin non-sensitivity” have increasingly been recognised in the literature, as there remains a significant proportion of aspirin-treated patients who experience recurrent vascular events. However, there is no consensus on the definition for aspirin resistance, which ranges from being defined as a clinical entity (thrombotic event while on aspirin treatment) to abnormalities of a range of biomarkers and enhanced platelet function testing. Although many embrace this concept, others believe that aspirin resistance may reflect treatment failure rather than “resistance” to aspirin. Aspirin is contraindicated in patients with a bleeding diathesis (as is the case with all the other groups of antiplatelet agents described below) and children under 16 years (risk of Reye’s syndrome).

Four adenosine diphosphate (ADP) P2Y12 receptor antagonists are licensed for clinical use, namely the thienopyridines clopidogrel and prasugrel; ticagrelor, which is a triazolopryrimidine; and cangrelor, an adenosine triphosphate (ATP) analogue. The former four are oral agents, whereas cangrelor is an intravenous antiplatelet agent (Table 1). Clopidogrel and prasugrel are prodrugs that require metabolism by the cytochrome P450 hepatic enzyme to become active [10]. Some patients have mutations in their CYP2C19 and CYP3A4 genes, which can lead to a decrease in the efficacy of clopidogrel. Laboratory genetic testing is cumbersome and not readily available as point-of-care testing, but can be tailored to high-risk bleeding patients who can have shorter durations of dual antiplatelet therapy (DAPT) [10]. Prasugrel has a faster onset of action and is a more potent P2Y12 inhibitor than clopidogrel. Prasugrel is not impacted by any genetic pleomorphism of the CYP2C19 gene [10]. The current recommended dose of prasugrel is a loading dose of 60 mg followed by a maintenance dose of 10 mg once daily, and in patients with low body weight or age > 75 the maintenance dose is reduced to 5 mg once daily. Prasugrel is contraindicated in patients with a history of previous stroke or (transient ischaemic attack) TIA. Ticagrelor, on the other hand, is already in an active form and acts as a direct and reversible P2Y12 inhibitor [10]. It has similar potency and similar onset of action as prasugrel, but it is administered twice daily. It inhibits the cellular uptake of adenosine, can cause dyspnoea in some patients and is contraindicated in patients with a history of intracranial haemorrhage. Cangrelor reversibly blocks the ADP P2Y12 receptor to achieve very potent platelet inhibition within a few minutes of its intravenous administration. It has a very short half-life of three to five minutes. On top of its very fast onset of action, cangrelor also has a fast off-set of 30 to 60 min for platelet aggregation to restore baseline levels [11] (Table 1). Cangrelor is also contraindicated in patients with a previous history of stroke or TIA.

The third group of antiplatelet agents available for clinical use are the glycoprotein IIb/IIIa inhibitors abciximab, tirofiban and eptifibatide, which are available for clinical use (Table 1). They are administered intravenously and target the final pathway of platelet aggregation, inhibiting the binding of GPIIb/IIIa with fibrinogen, von Willebrand and other ligands. Abciximab has a high affinity for the fibrinogen receptors, and has a long platelet-bound half-life. Dose adjustment is not required in the setting of renal impairment. On the other hand, tirofiban and eptifibatide have shorter platelet-bound half-lives. They are both renally excreted, and therefore require dose adjustment in renal impairment [12]. Glycoprotein IIb/IIIa inhibitors are contraindicated in patients with haemorrhagic stroke within the last 30 days. 

Another class of antiplatelet agent is the phosphodiesterase inhibitor dipyridamole. It increases intracellular cyclic adenosine monophosphate (cAMP) and cyclic guanosine monophosphate (cGMP) levels in platelets and smooth muscle cells [13]. This results in the inhibition of platelet aggregation and vasodilation of coronary arteries. Dipyridamole is rapidly absorbed after oral administration, and reaches its peak plasma concentration within two hours. It has a half-life of approximately 10 h and is primarily metabolized in the liver. Dipyridamole and its metabolites are mainly excreted in the urine [13]. Its common side effects include angina pectoris, diarrhoea, dizziness, headache, myalgia, nausea and vomiting, and skin reactions. 

Lastly, vorapaxar is a thrombin receptor inhibitor that prevents thrombin from activating platelets via the protease-activated receptor (PAR)-1 [14]. Vorapaxar is rapidly absorbed after oral administration, and reaches its peak plasma concentration within two to four hours. It has a half-life of approximately five days, and is primarily metabolised in the liver by cytochrome P450 enzymes [15]. It is recommended for use in combination with aspirin or clopidogrel, and contraindicated in patients with a history of stroke, TIA or bleeding disorders [16].

## 3. Antiplatelet Therapy in Primary Prevention for Atherosclerotic Cardiovascular Disease (ASCVD)

Aspirin has been widely used for decades for primary prevention of ASCVD. In low and middle income countries, studies such as the PolyIran study [17] have shown that aspirin-containing polypill strategies were effective in preventing major cardiovascular events. A large meta-analysis by the Antithrombotic Trialists’ Collaboration in 2009 showed aspirin to have a small benefit in reducing serious vascular events, but at the expense of an increase in gastrointestinal and extracranial bleeding in primary prevention trials [18]. More recent primary prevention trials conducted targeting the elderly (ASPREE trial [19]), individuals with diabetes mellitus (ASCEND trial [20]), and those at moderate risk of a cardiovascular event (ARRIVE trial [20]) also confirmed that the benefit of aspirin was marginal at best, whilst posing a major bleeding hazard in participants who were otherwise healthy. As a result, the 2021 European Society of Cardiology (ESC) guideline [21] does not recommend antiplatelet therapy for individuals at low or moderate risk of ASCVD due to the higher risk of major bleeding. However, in patients with diabetes mellitus at high or very high ASCVD risk, low-dose aspirin may be considered for primary prevention (Class IIb, Level A). Likewise, the American College of Cardiology/American Heart Association (ACC/AHA) guidelines only recommend low-dose aspirin in individuals aged 40 to 70 years who are at increased risk of ASCVD and low risk of bleeding (Class IIb, Level A) whereas it is not recommended among those aged >70 years or at increased risk of bleeding [22].

## 4. Antiplatelet Therapy Following Elective Percutaneous Coronary Intervention (PCI) for Chronic Coronary Syndrome (CCS)

For patients with stable coronary artery disease undergoing elective PCI for stable angina, the preferred dual antiplatelet therapy (DAPT) of choice is aspirin and clopidogrel, as shown in the landmark CREDO trial [23]. The duration of DAPT following elective PCI in patients with CCS has been widely studied. The latest ESC guidelines [24] recommend six months of DAPT, whereas the ACC/AHA guidelines [25] recommend three to six months of DAPT following elective PCI (EXCELLENT [26], SECURITY [27] AND ISAR-SAFE [28] trials). The benefit beyond this period is minimal at the expense of an increased risk of major bleeding [29,30] and non-cardiovascular death [31,32,33], as shown in several meta-analyses. A shorter duration of dual antiplatelet medication (between one and three months) can be considered in patients with high bleeding risk (GLOBAL LEADERS [34], MASTER DAPT [35], STOPDAPT 2 [36] and one-month DAPT [37] trials). Therefore, some patients would benefit from an individualised approach and tools such as PRECISE-DAPT [7] and DAPT [6] to risk-stratify patients and aid clinicians in identifying those at high ASCVD risk who may benefit from a prolonged duration of DAPT, and for those with high bleeding risk who would warrant a shorter duration of DAPT.

In terms of intravenous antiplatelet agents, there currently is no role for glycoprotein IIb/IIIa inhibitors in CCS patients undergoing PCI [25]. However, cangrelor may be considered in P2Y12 inhibitor-naive patients undergoing PCI (CHAMPION PHOENIX [38]), and is supported by the ESC (Class IIb, Level B) [39] and ACC/AHA (Class IIb, Level B) [25] guidelines.

## 5. Antiplatelet Therapy in Acute Coronary Syndrome (ACS)

Following the CURE [40] and CLARITY-TIMI 28 [41] studies, DAPT with aspirin and clopidogrel became the cornerstone in the management of ACS for more than a decade. However, following the publication of the PLATO [3] and TRITON-TIMI 38 [2] trials, the landscape of DAPT choice in the management of ACS has changed. Ticagrelor became the antiplatelet of choice alongside aspirin for the management of unstable angina and non-ST-segment elevation myocardial infarction (NSTEMI). Moreover, among ACS patients with high bleeding risk, the TWILIGHT trial [42] demonstrated that 3 months of DAPT with aspirin and ticagrelor after PCI was associated with fewer bleeds than 12 months of DAPT, without increasing the risk of ischaemic events. Prasugrel, on the other hand, was not superior to clopidogrel in medically managed NSTEMI (TRILOGY-ACS trial) [43] and pre-treatment with prasugrel in NSTEMI patients awaiting coronary angiography was also found not to be beneficial in reducing ischaemic endpoints but increased bleeding complications (ACCOAST trial [44]). Prasugrel was compared head-to-head with ticagrelor in the ISAR-REACT 5 trial [45] in NSTEMI and STEMI patients. Prasugrel was found to be superior to ticagrelor in reducing the composite endpoint of death, myocardial infarction and stroke, with no difference in major bleeding. As a result, one year of DAPT with aspirin and prasugrel is now recommended over aspirin and ticagrelor in STEMI and NSTEMI patients (without prior history of stroke or TIA) undergoing PCI by current ESC [46] and ACC/AHA [25] guidelines.

The intravenous glycoprotein IIb/IIIa inhibitors (abxicimab, tirofiban, eptifibatide) are not routinely recommended for use in ACS patients, due to a lack of consistent benefit in previous trials (CADILLAC, ISAR-REACT-4 [47,48]). Currently, they are only recommended for use in ACS patients undergoing PCI in the setting of high thrombus burden, slow flow or no-reflow [25,49] (Class IIa, Level C). As with patients with CCS, cangrelor may be considered in P2Y12 inhibitor-naive ACS patients undergoing PCI [25,39].

## 6. Role of Antiplatelets Post-Coronary Artery Bypass Graft (CABG)

In patients who have undergone CABG, the patency of saphenous vein grafts (SVGs) varies over time, with up to 40% of saphenous venous grafts (SVGs) and 15% of internal mammary arteries being occluded at 10 years [50]. Historically, aspirin administration in the perioperative period was reported to have improved mortality, and a large metanalysis comparing SVG patency in post-CABG patients reported a higher graft patency in patients taking moderate doses of aspirin (325 mg) when compared to low dose aspirin (50–100 mg). 

Prior trials that looked at graft patency with DAPT when compared with aspirin monotherapy in CABG patients were small and underpowered. This was recently summarised in a meta-analysis of five randomised controlled trials (*n* = 837), and DAPT did not significantly reduce graft occlusion [51]. A prior meta-analysis, which also included non-randomised studies, showed that in studies involving off-pump CABG, DAPT reduced the graft occlusion rate by 55% [52]. Recently, the POPular CABG trial compared DAPT with aspirin and ticagrelor against aspirin monotherapy in 499 patients who underwent CABG, and at one year the SVG occlusion rate was similar in both groups [53]. However, the most recent meta-analysis of four randomised clinical trials [53,54,55,56,57], including the POPular CABG trial [53], concluded that adding ticagrelor to aspirin decreased the risk of SVG failure, but was associated with increased bleeding risk [58].

The current ESC [39] and AHA [59] guidelines recommend DAPT with aspirin and a PY12 inhibitor in all ACS patients who undergo CABG (with transient interruption in the peri-operative period). In those with CCS undergoing CABG, low-dose aspirin monotherapy in the peri-operative period and long-term is recommended by the ESC guidelines (Class I, Level C) [39]. In those perceived to be at a high ischaemic risk with prior MI and CABG, DAPT for 12 months and up to 36 months may be considered (Class IIb, Level C). However, the AHA guideline varies for CCS patients depending on whether they had on-pump or off-pump CABG, with 1-year DAPT with aspirin and ticagrelor recommended for the latter (Class 1, Level A) [59]. When aspirin monotherapy is being considered, a higher dose of 325 mg rather than the lower dose of 81 mg is preferred (Class IIa, Level A).

## 7. Long-Term Antiplatelet Therapy for Secondary Prevention in IHD Patients

In patients with established ASCVD, long-term low-dose aspirin monotherapy has so far been the antiplatelet agent of choice to prevent adverse outcomes based on the results of the landmark meta-analysis from the Antithrombotic Trialists’ Collaboration [18]. However, several trials (summarised in a recent meta-analysis [60]) have recently compared aspirin monotherapy against P2Y12 inhibitor monotherapy, and the latter has been shown to be superior to aspirin in reducing MI, with a similar bleeding risk. Of note, the number needed to treat to prevent one myocardial infarction was high at 244. Longer-term follow-up results are starting to emerge as recently reported by the HOST-EXAM Extended study [61], and future guidelines are likely to be influenced by these results.

Based on the promising results in a subgroup of patients in the CHARISMA trial [62], the role of prolonged DAPT with ticagrelor and aspirin for secondary prevention in high-risk stable patients (age > 65 years, chronic kidney disease, second presentation of MI, multi vessel coronary artery disease and diabetes mellitus on treatment) with a history of prior MI were investigated in the PEGASUS—TIMI 54 trial [63]. After a median follow-up of nearly three years, prolonged DAPT was associated with significantly fewer ischaemic events, but with more bleeding events than aspirin monotherapy alone. Furthermore, low-dose ticagrelor (60 mg BD) was associated with significantly fewer side effects. Therefore, there may be a subset of patients who would benefit from three years of DAPT with low-dose ticagrelor and aspirin, based on their high-risk features. Further risk stratification may be achieved with the addition of scores generated by utilising risk stratification tools such as PRECISE-DAPT [7] and DAPT [6].

## 8. Role of Antiplatelets in Post TAVI

Thromboembolic complications following TAVI include obstructive valve thrombosis [64], peri-procedural MI [65] and stroke [66], for example, due to interaction between the transcatheter valve system and diseased aorta/aortic valve [67]. Patient characteristics can also increase the risk of thromboembolic events after TAVI, including older age [68], left ventricular dysfunction [69,70], atrial fibrillation [71], prior cerebrovascular events [68], obesity and chronic kidney disease [72].

POPULAR TAVI [73] is currently the largest randomised trial that compared single antiplatelet (SAPT—aspirin only) vs. DAPT (aspirin + clopidogrel for three months) post TAVI, and it showed an increase in bleeding events in the DAPT group when compared to the SAPT group. There was no significant reduction in cardiovascular death or stroke. Likewise, the ARTE trial [74] compared SAPT vs. DAPT post TAVI, and showed that major life-threatening bleeding and vascular complications were higher in DAPT group when compared to SAPT group; moreover, there was no significant difference in all causes of death and stroke between both groups. 

Therefore, the ESC guideline recommends lifelong SAPT (preferably with aspirin, but clopidogrel is an alternative) in patients without concurrent indication for chronic oral anticoagulation (OAC) and without recent coronary stents (class 1, level of evidence A) [75]. 

In patients who have coronary stenting within 3 months of TAVI, and no pre-existing indication for OAC, the ESC recommends DAPT with aspirin and clopidogrel for 1–6 months, followed by SAPT [76]. A longer duration of DAPT is not recommended in this setting, because TAVI patients tend to have a higher underlying bleeding risk due to comorbidities [76].

In patients undergoing TAVIs with pre-existing indications for OAC, the ESC recommends OAC alone long-term (class 1, level of evidence B) [75]. This recommendation is based on evidence from the POPULAR TAVI (cohort B) randomised trial, which found a lower incidence of bleeds when OAC alone was compared to OAC plus clopidogrel [73]. However, in patients with concurrent indication for chronic OAC who have had coronary stenting within 3 months of TAVI, a single antiplatelet drug for 1–6 months is recommended along with long-term OAC [24].

## 9. Role of Antiplatelets in Non-Cardioembolic Ischaemic Stroke and TIAs

Aspirin is primarily used as secondary prevention in patients with TIA and stroke, as it has been shown to reduce the risk of recurrence by 22% [77]. Extended-release dipyridamole in combination with aspirin showed some promise to reduce the recurrence of adverse events in patients with ischaemic stroke (ESPRIT trial) [78]. However, the much larger PRoFESS trial [79] failed to demonstrate a reduction in stroke recurrence, but revealed an increased risk of major bleeding with extended-release dipyridamole and aspirin when compared to aspirin alone. Based on these trials, DAPT with extended-release dipyridamole and aspirin is not currently used in stroke prevention or TIAs in patients with recent ischaemic strokes.

A prolonged duration of DAPT with aspirin (325 mg) and clopidogrel (75 mg) in patients with lacunar stroke did not reduce recurrent stroke but showed an increase in major bleeding and death in the SPS3 trial [80]. However, short-term DAPT with low-dose aspirin and clopidogrel in patients with minor stroke or high-risk TIAs has been compared in the CHANCE [81] and POINT [82] trials; a pooled analysis of the two trials [83] showed that DAPT with low-dose aspirin and clopidogrel was superior to low-dose aspirin in reducing composite ischaemic events, the benefits of which were mainly seen within the first 21 days of the index event.

DAPT with low-dose aspirin and ticagrelor for 30 days compared to low-dose aspirin monotherapy in patients with mild to moderate stroke or TIA has been evaluated in the THALES trial [84]. Low-dose aspirin and ticagrelor significantly reduced the 30-day composite ischaemic endpoints, but also increased the incidence of severe bleeding, including intracranial bleeding.

Short-term low-dose aspirin monotherapy has also been compared with ticagrelor monotherapy in non-severe stroke and high-risk TIA patients in the SOCRATES trial [85]. In this setting, ticagrelor was not found to be inferior to aspirin in the prevention of recurrent ischaemic endpoints without an increased risk of major bleeding.

Based on these trials, the current American and European guidelines [86,87] recommend DAPT (aspirin with clopidogrel) for the first 90 days followed by long-term aspirin in high-risk TIA and in early non-cardioembolic moderate to severe ischaemic stroke patients. In low-risk TIA and in late non-cardioembolic ischaemic stroke patients, single antiplatelet therapy is recommended.

## 10. Antiplatelets in Peripheral Artery Disease (PAD)

The prevalence of concomitant cardiovascular and cerebrovascular disease is high in patients suffering from PAD, as they share a similar risk profile. Primary prevention with aspirin in patients with asymptomatic PAD did not demonstrate any clinical benefit in the absence of concomitant cardiovascular and cerebrovascular disease (Aspirin for Asymptomatic Atherosclerosis Trialists trial) [88]. Hence, antiplatelet therapy is generally not indicated for primary prevention in asymptomatic PAD in the absence of concomitant cardiovascular disease.

Symptomatic patients with PAD can be further divided into two categories. The first category includes patients with symptomatic PAD with co-existing cardiovascular disease. In these patients who have not undergone any recent intervention, current evidence suggests that the concomitant use of low-dose aspirin and low-dose rivaroxaban (2.5 mg BD) was superior to aspirin monotherapy in reducing cardiovascular death and non-fatal stroke, but was associated with an increase in major bleeding (COMPASS trial) [89]. A subsequent analysis of the net clinical benefit (defined as the composite of cardiovascular death, stroke, MI, fatal bleeding, or symptomatic bleeding into a critical organ) from the COMPASS trial [89] confirmed the superiority of aspirin and low-dose rivaroxaban in these patients, and that this benefit was more pronounced in a subset of high-risk patients (those with ASCVD in at least two vascular beds, impaired renal function, heart failure or diabetes) [90]. The second category of patients are those with symptomatic PAD and with no cardiovascular disease. In this cohort, antiplatelet therapy with clopidogrel has been shown to be more effective than aspirin, as demonstrated in the CAPRIE trial [91], whereas ticagrelor was not found to be superior to clopidogrel in reducing cardiovascular events in a similar population, as shown in the EUCLID trial [92].

In symptomatic PAD patients who have undergone lower limb revascularisation, long-term low-dose aspirin and low-dose rivaroxaban (2.5 mg BD) was associated with a significant reduction in ischaemic endpoints when compared to aspirin monotherapy (VOYAGER PAD trial) [93].

Vorapaxar was also evaluated in the TRA 2°P-TIMI 50 trial [14], which included patients with established symptomatic ASCVD. Within the subgroup of patients with PAD [5], vorapaxar reduced acute limb ischaemia and peripheral revascularisation. However, this benefit was offset by an increased risk of bleeding when compared to routine care. 

Therefore, based on these studies, current American [94] and European [95] guidelines recommend single antiplatelet therapy (preferably with clopidogrel) in patients with symptomatic PAD (Class 1, Level A). Short-term DAPT with aspirin and clopidogrel could be considered in those who undergo revascularisation (Class IIb, Level B). In patients with asymptomatic PAD, the guidelines differ with the American guideline [94] that recommends SAPT to reduce adverse ischaemic events, and the European guideline [95] that recommends aspirin in those with asymptomatic >50% carotid artery stenosis, but not in those with asymptomatic lower extremity arterial disease. Of note, both these guidelines pre-date the publication of the COMPASS trial [89,90] and it is very likely that future updates of these guidelines will recommend long-term aspirin with low-dose rivaroxaban in high-risk patients with low bleeding risk. 

## 11. Antiplatelet Therapy in COVID-19

COVID-19 infection is recognised to be associated with both acute-phase arterial and venous thromboses [96]. This is primarily due to endothelial injury, inflammation and impaired coagulation. Platelets play a pivotal role in haemostasis and thrombus formation. The complex role of platelets in immune response-related thrombus formation has recently been implicated in the thrombotic phenomena observed with COVID-19 infections [97]. As a result, the role of antiplatelet therapy with aspirin and PY12 inhibitors has recently gathered significant interest. 

Recently, several trials have been reported in this setting. The ACTIV 4a trial [98] compared the effect of adding a PY12 inhibitor to anticoagulation in non-critically ill patients with COVID-19. This combination therapy did not impact on clinical outcomes after 21 days of follow-up when compared to anticoagulation alone. The ACTIV 4b trial [99] compared the use of low-dose aspirin with low-dose apixaban (2.5 mg BD), apixaban alone (5 mg BD) and a placebo in clinically stable but symptomatic COVID-19 patients in an outpatient setting. This trial was terminated early due to low event rates across all groups, and there was no evidence of benefit with the use of antiplatelet or anticoagulation in this group of patients. The RECOVERY trial [100] compared the impact of adding aspirin 150 mg to standard care in patients hospitalised with COVID-19. Aspirin did not reduce 28-day mortality or progression to requiring invasive mechanical ventilation. The REMAP-CAP trial [101] enrolled critically ill patients with COVID 19 and randomised them to either aspirin, a P2Y12 inhibitor versus no antiplatelets in addition to thromboprophylaxis. After a follow-up of 21 days, there was no difference in the number of organ support-free days between the two groups. The COVID-PACT trial [102] was a smaller trial that compared full-dose anticoagulation against clopidogrel in a 2 × 2 factorial design in critically-ill COVID-19 patients. In this setting, clopidogrel did not reduce thrombotic complications within the first 28 days, whereas full-dose anticoagulation did demonstrate a reduction.

Therefore, currently there is no evidence to suggest that antiplatelet therapy is beneficial in symptomatic COVID-19 patients in the acute phase. However, several other trials are currently underway, as summarised by Talasaz et al. [103] in a recent comprehensive review, and their results are awaited.

## 12. Conclusions

Antiplatelet therapy plays a crucial role in a range of cardiovascular diseases. In this review, we provided an overview of the latest evidence and recommendations for antiplatelet therapy in various clinical settings, ranging from primary prevention, acute phase of ACS, stroke and COVID-19 infection, its peri-procedural use in PCI, to TAVI, to secondary prevention in IHD and PAD (Figure 1). We currently have a range of antiplatelet agents in our armamentarium to improve clinical outcomes, supported by evidence from robust trials. Together with the use of clinical decision-making tools and risk predictors, we can now tailor therapies for individualised care.

## Figures and Tables

**Figure 1 ijms-24-11132-f001:**
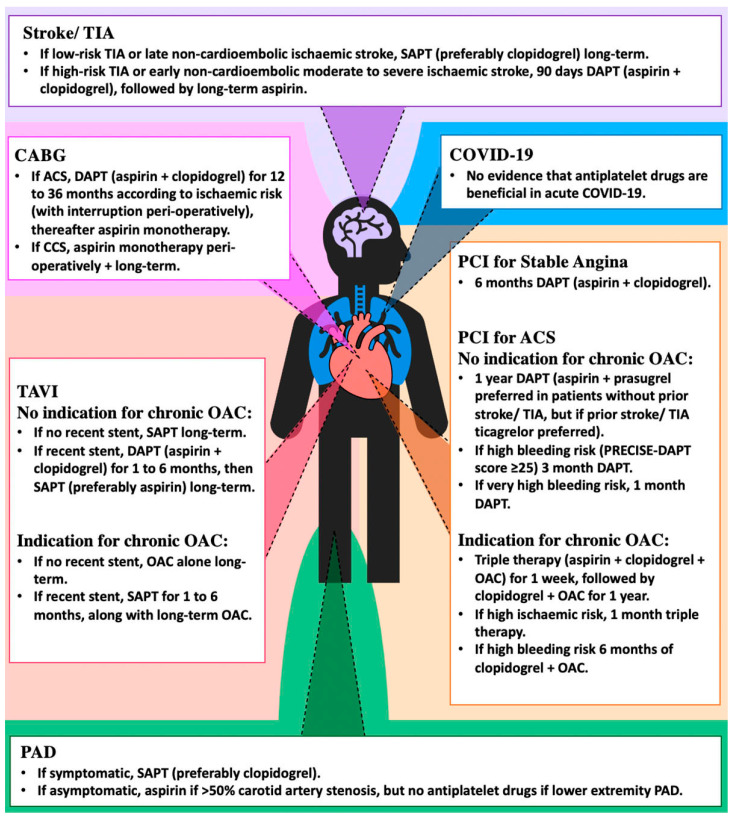
Indications of antiplatelet agents in various clinical settings.

**Table 1 ijms-24-11132-t001:** Antiplatelet drugs in clinical use.

Drug	Structure	Mechanism	Dose	Preparation	Onset of Action	Indications
Aspirin	Acetylsalicylic acid	COX-1 inhibitor	300 mg loading 75–100 mg maintenance	Oral/PR	<60 min	ACS, TIA, ischaemic stroke, CABG, PVD, PAD, coronary stents
Clopidogrel	Thienopyridine	ADP P2Y12 inhibitor	300–600 mg loading 75 mg maintenance	Oral	2–8 h	ACS, ischaemic stroke, PAD, PVD, coronary stents
Prasugrel	Thienopyridine	ADP P2Y12 inhibitor	60 mg loading 5–10 mg maintenance	Oral	30 min–4 h	ACS
Ticagrelor	Triazolopyrimidine	ADP P2Y12 inhibitor	180 mg loading 90 mg BD maintenance	Oral	30 min–4 h	ACS
Cangrelor	ATP analogue	ADP P2Y12 inhibitor	30 mcg/kg bolus loading 4 mcg/kg/min maintenance *	Intravenous	2 min	ACS
Abciximab	human-murine chimeric monoclonal antibody fragment	glycoprotein IIb/IIIa inhibitor	0.25 mg/kg bolus loading 0.125 mcg/kg/min maintenance 12 h	Intravenous	<10 min	ACS, PCI
Tirofiban	Non-peptide mimetic	glycoprotein IIb/IIIa inhibitor	25 mcg/kg bolus loading 0.15 mcg/kg/min maintenance <18 h	Intravenous	<10 min	ACS, PCI
Eptifibatide	Synthetic cyclic heptapeptide	glycoprotein IIb/IIIa inhibitor	180 mcg/kg double bolus loading 2 mcg/kg/min maintenance <18 h	Intravenous	<15 min	ACS, PCI
Vorapaxar	Tricyclic himbacine derivative	PAR-1 inhibitor	2.08 mg maintenance	Oral	<1 week	ACS, PAD
Dipyridamole	Dialkylarylamines	Phosphodiesterase inhibitor	modified release: 200 mg twice a day Immediate release: 300–600 mg in divided doses	Oral	24 min	ischaemic stroke

* Continue for 2 h or duration of the procedure, whichever is longer. ATP: adenosine triphosphate, ADP: adenosine diphosphate, PAR: protease-activated receptor, ACS: acute coronary syndrome, PAD: peripheral artery disease, PVD: peripheral vascular disease, TIA: transient ischaemic attack, CABG: coronary artery bypass graft, PCI: primary cutaneous intervention, PR: per rectal.

## Data Availability

Not applicable.

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
