# Peer review of "A Contemporary Review of Antiplatelet Therapies in Current Clinical Practice"

_ijms, 2023, doi:10.3390/ijms241311132_

Round 1
Reviewer 1 Report
Overall this is a nice well written review. I have some minor comments
1) It would be good to acknowledge the effect of different aspirin doses on not only TXA2 but also PGI2 levels, given this could be one reason why doses may have differing effects dependent on the clinical context.
2) Please add 'by' into 'Prasugrel is not impacted any genetic pleomorphism of the CYP2C19 gene[10]'
3) Please ensure DAPT abbreviation is explained when first used (on page 5)
4) Check TWILIGHT spelling
5) please correct this sentence 'In terms of intravenous antiplatelet agents, there currently s no role for glycoprotein IIbIIIa inhibitors in CCS patients undergoing PCI[25].'
6) Please relook at this sentence it does not at present make sense in terms of the English. 'POPULAR TAVI is currently the largest randomised trial compared single anti platelet (SAPT) vs DAPT (aspirin only vs aspirin + clopidogrel for three months) post TAVI showed increase in bleeding in DAPT group vs aspirin only group.'
This review is well written. There are just some very minor editing issues as indicated above.
Author Response
Overall this is a nice well written review. I have some minor comments
1) It would be good to acknowledge the effect of different aspirin doses on not only TXA2 but also PGI2 levels, given this could be one reason why doses may have differing effects dependent on the clinical context.
We thank the reviewer for this comment. We have added the following sentences in this section to this effect:
“Aspirin has also been shown to block prostaglandins (e.g. PGE2 and PGI2), which has a proinflammatory effect. Earlier researchers found that lower doses of aspirin had greater inhibitory effects on thromboxane A2 than PGI2 metabolites. However, they found that platelet function inhibition was not maximal at the lower aspirin doses and that aspirin at daily doses > 80 mg caused substantial inhibition of endogenous PGI2 production.”
2) Please add 'by' into 'Prasugrel is not impacted any genetic pleomorphism of the CYP2C19 gene[10]'
We thank the reviewer for highlighting this typo. We have now amended this section accordingly.
3) Please ensure DAPT abbreviation is explained when first used (on page 5)
We thank the reviewer for highlighting this omission. We have now explained DPAT when first used.
4) Check TWILIGHT spelling
We thank the reviewer for highlighting this typo. We have now amended this section accordingly.
5) please correct this sentence 'In terms of intravenous antiplatelet agents, there currently s no role for glycoprotein IIbIIIa inhibitors in CCS patients undergoing PCI[25].'
We thank the reviewer for highlighting this typo. We have now amended this section accordingly.
6) Please relook at this sentence it does not at present make sense in terms of the English. 'POPULAR TAVI is currently the largest randomised trial compared single anti platelet (SAPT) vs DAPT (aspirin only vs aspirin + clopidogrel for three months) post TAVI showed increase in bleeding in DAPT group vs aspirin only group.'
We thank the reviewer for highlighting this sentence. We have reworded it as below:
“POPULAR TAVI is currently the largest randomised trial that compared single antiplatelet (SAPT – aspirin only) vs DAPT (aspirin + clopidogrel for three months) post TAVI and it showed an increase in bleeding events in the DAPT group when compared to the SAPT group.”
Reviewer 2 Report
Summary
This manuscript provides an extensive review of antiplatelet medications (dose, MOA, indications) and their current use in clinical practice. In particular, the authors discuss antiplatelet use in ASCVD, TAVI, and COVID-19. The authors have submitted a well written manuscript that provides a strong summary of antiplatelet therapy and clinical guidelines.
One of the areas I feel this paper can be improved is the identification and discussion of knowledge gaps in the areas of clinical practice discussed. This would add more novelty to the paper above a general summarization of the current guidelines, which I felt was done very well by the authors. Please see Major point one for an example of one such change. Is there any opportunity for the authors to discuss future directions or areas of study (i.e. clinical trials to address current knowledge caps)?
In terms of completeness of the topic covered, the paper is quite thorough. There is the possibility of discussing the role of aspirin as an anticancer agent. This has become an area of interesting research in recent years. It may also be possible to discuss how clinicians can go about resuming antiplatelet agents after a major bleed. Other indications for aspirin like Kawasaki’s disease, essential thrombocytosis, etc. are not discussed.
Minor Suggestions
1. Table 1, not enough space between column 2 “structure” and column 3 “mechanism. As a result, “monoclonal” nearly touches “glycoprotein” in the abciximab row. This occurs also in the columns labelled “dose”, “preparation” and “onset of action”. Please widen the columns to improve spacing.
2. Table 1, Aspirin dose range maybe 75-150 mg in some contexts, please double check this.
3. Table 1, Dipyramidole is cut off taking up two lines. Please change width to fit properly.
4. Table 1, ADP is written in long form many times. Please just use “ADP” in the mechanism column for clopidigrel through to cangrelor. Then define “ADP” as done for other acronyms in table legend. Ensure table legend for acronyms is in line with the table.
5. Please change to have inhibiting instead of blocking: “Aspirin is the most widely used antiplatelet agent and exerts its antiplatelet property by irreversibly blocking INHIBITING the cyclooxygenase-1 (COX-1) enzyme present in the platelets.”
6. Section 2 paragraph 3. Please change “off-loading dose” to “loading dose”. I have never heard of this terminology before.
7. Section 2 paragraph 3. “Cangrelor reversibly blocks the ADP receptor to…” Please change to specify the receptor “Cangrelor reversibly blocks the ADP P2Y12 receptor to…”
8. Section 2 paragraph 3. Please fix this sentence it is vague and grammatically incorrect. “On top of it very fast onset of action, it also has a fast off-set…”. I suggest: “On top of its very fast onset of action, prasugrel also has a fast off-set…”
9. Section 2. Please briefly try to improve discussion of key side effects and contraindications for each class
10. There were quite a few typos throughout, not all were caught by spell check.
Major Suggestions
1. Section 2 paragraph 2 as well as Section 3. Please comment on aspirin nonsensitivity/resistance in the discussion of aspirin in the manuscript as it is a well studied issue that limits efficacy for a significant proportion of patients. There have been cases of recurrent thrombosis, especially stent thrombosis, despite being treated with these agents. Since you discuss the pharmacogenetics of clopidogrel, I feel that this would improve the brief paragraph on aspirin or add some content that is unknown in the field. Below is a recent paper on the topic:
Aspirin nonsensitivity in patients with vascular disease: Assessment by light transmission aggregometry (aspirin nonsensitivity in vascular patients) RPTH. Hamzah Khan et al.
2. Section 2 paragraph 4. “They are administered intravenously and target the final pathway of platelet aggregation by competing with fibrinogen and von Willebrand factor to bind IIbIIIa receptors.” There are a few issues with this sentence listed below:
a. It should be written “GPIIbIIIa” and not “IIbIIIa”
b. More importantly this sentence should be updated to reflect the current understanding of this receptor’s multiple and complex ligand interactions by saying something like: “They are administered intravenously and target the final pathway of platelet aggregation by inhibiting the binding of GPIIbIIIa with fibrinogen, von Willebrand, and other ligands.”
Persistence of platelet thrombus formation in arterioles of mice lacking both von Willebrand factor and fibrinogen. JCI
Heyu Ni, Cécile V. Denis, Sangeetha Subbarao, Jay L. Degen, Thomas N. Sato, Richard O. Hynes, and Denisa D. Wagner
Several minor typos, otherwise well written manuscript.
Author Response
Minor Suggestions
- Table 1, not enough space between column 2 “structure” and column 3 “mechanism. As a result, “monoclonal” nearly touches “glycoprotein” in the abciximab row. This occurs also in the columns labelled “dose”, “preparation” and “onset of action”. Please widen the columns to improve spacing.
We thank the reviewer for this comment. We have now widened the columns accordingly
- Table 1, Aspirin dose range maybe 75-150 mg in some contexts, please double check this.
We thank the reviewer for this comment. 150mg of Aspirin was used in the RECOVERY trial for covid patients. However, this did not improve clinical outcomes. Therefore we have kept the most common dose of aspirin at 75-100mg.
- Table 1, Dipyramidole is cut off taking up two lines. Please change width to fit properly.
We thank the reviewer for this comment. We have now widened the columns accordingly to fix this issue
- Table 1, ADP is written in long form many times. Please just use “ADP” in the mechanism column for clopidigrel through to cangrelor. Then define “ADP” as done for other acronyms in table legend. Ensure table legend for acronyms is in line with the table.
We thank the reviewer for this comment. We have now changed this to ADP in the table and defined it a the end of the table.
- Please change to have inhibiting instead of blocking: “Aspirin is the most widely used antiplatelet agent and exerts its antiplatelet property by irreversibly blocking INHIBITING the cyclooxygenase-1 (COX-1) enzyme present in the platelets.”
We thank the reviewer for this suggestion. We have changed it accordingly.
- Section 2 paragraph 3. Please change “off-loading dose” to “loading dose”. I have never heard of this terminology before.
We thank the reviewer for this comment. We have amended it accordingly.
- Section 2 paragraph 3. “Cangrelor reversibly blocks the ADP receptor to…” Please change to specify the receptor “Cangrelor reversibly blocks the ADP P2Y12 receptor to…”
We thank the reviewer for this comment. We have amended this sentence accordingly.
- Section 2 paragraph 3. Please fix this sentence it is vague and grammatically incorrect. “On top of it very fast onset of action, it also has a fast off-set…”. I suggest: “On top of its very fast onset of action, prasugrel also has a fast off-set…”
We thank the reviewer for this comment. We have amended this sentence accordingly (cangrelor instead of prasugrel)
- Section 2. Please briefly try to improve discussion of key side effects and contraindications for each class
We thank the reviewer for this comment. We have improved the discussion of key contraindications and side-effects where applicable.
- There were quite a few typos throughout, not all were caught by spell check.
Major Suggestions
- Section 2 paragraph 2 as well as Section 3. Please comment on aspirin nonsensitivity/resistance in the discussion of aspirin in the manuscript as it is a well studied issue that limits efficacy for a significant proportion of patients. There have been cases of recurrent thrombosis, especially stent thrombosis, despite being treated with these agents. Since you discuss the pharmacogenetics of clopidogrel, I feel that this would improve the brief paragraph on aspirin or add some content that is unknown in the field. Below is a recent paper on the topic:
Aspirin nonsensitivity in patients with vascular disease: Assessment by light transmission aggregometry (aspirin nonsensitivity in vascular patients) RPTH. Hamzah Khan et al.
We thank the reviewer for this suggestion. We have added the following comments to this effect and referenced this study suggested:
“The terms “aspirin resistance” or aspirin non-sensitivity” have increasingly been recognised in the literature as there remains a significant proportion of aspirin-treated patients (up to 24% in a recent study (ref)) who experiences recurrent vascular events. However, there is no consensus on the definition for aspirin resistance which ranges from being defined as a clinical entity (thrombotic event while on aspirin treatment) to abnormalities of a range of biomarkers and enhanced platelet function testing. Although many embrace this concept, others believe that aspirin resistance may reflect treatment failure rather than “resistance” to aspirin. Aspirin is contraindicated in patients with a bleeding diathesis (as is the case with all the other groups of antiplatelet agents described below) and children under 16 years (risk of Reye’s syndrome).”
- Section 2 paragraph 4. “They are administered intravenously and target the final pathway of platelet aggregation by competing with fibrinogen and von Willebrand factor to bind IIbIIIa receptors.” There are a few issues with this sentence listed below:
- It should be written “GPIIbIIIa” and not “IIbIIIa”
- More importantly this sentence should be updated to reflect the current understanding of this receptor’s multiple and complex ligand interactions by saying something like: “They are administered intravenously and target the final pathway of platelet aggregation by inhibiting the binding of GPIIbIIIa with fibrinogen, von Willebrand, and other ligands.”
Persistence of platelet thrombus formation in arterioles of mice lacking both von Willebrand factor and fibrinogen. JCI
Heyu Ni, Cécile V. Denis, Sangeetha Subbarao, Jay L. Degen, Thomas N. Sato, Richard O. Hynes, and Denisa D. Wagner
We thank the reviewer for thes suggestions. We have amended accordingly with updated references.